# An Ethanol Vapor Sensor Based on a Microfiber with a Quantum-Dot Gel Coating

**DOI:** 10.3390/s19020300

**Published:** 2019-01-14

**Authors:** Siqi Hu, Guofeng Yan, Chunzhou Wu, Sailing He

**Affiliations:** 1Centre for Optical and Electromagnetic Research, State Key Laboratory of Modern Optical Instrumentation, Zhejiang Provincial Key Laboratory for Sensing Technologies, Zhejiang University, Hangzhou 310058, China; siqihu@zju.edu.cn (S.H.); 21630092@zju.edu.cn (C.W.); sailing@zju.edu.cn (S.H.); 2Zhejiang Lab, Hangzhou 310000, China

**Keywords:** optical fiber ethanol sensor, microfiber, quantum-dot gel

## Abstract

An ethanol vapor sensor based on a microfiber with a quantum-dot (QD) gel coating is proposed and demonstrated. The QD gel was made from UV glue as the gel matrix and CdSe/ZnS QDs with a concentration of 1 mg/mL. The drawing and coating processes were conducted by using a simple and low-cost system developed for this study. Bending, ethanol sensing, temperature response, and time response tests were carried out, respectively. The experimental results showed that the fabricated sensor had a high sensitivity of −3.3%/ppm, a very low temperature cross-sensitivity of 0.17 ppm/°C, and a fast response time of 1.1 s. The easily fabricated robust structure and the excellent sensing performance render the sensor a promising platform for real ethanol sensing applications.

## 1. Introduction

Ethanol concentration is an important parameter in many applications ranging from biomedicine to safety, and is used even in our daily lives, for example, the alcohol test for drivers. Various ethanol sensors have been proposed and developed based on different sensing principles, such as electric gas sensors [1,2], SAW (surface acoustic wave) gas sensors [3], and optical fiber gas sensors [4,5,6,7,8,9,10]. Among them, optical fiber sensors have attracted more attention during these past years, due to their unique advantages of immunity to electromagnetic interference, high sensitivity, and compact size. For example, it has been proposed and demonstrated that a single-mode silica fiber Bragg grating (FBG) coated with a thin layer of poly (methyl methacrylate) (PMMA) exhibits good sensing properties for ethanol detection [4]. Similarly, an optical fiber long-period grating (LPG) with a ZnO nanorod coating was also developed for ethanol vapor sensing [5]. Recently, an ethanol gas sensor based on a hybrid PMMA-silica microfiber coupler was investigated and the experimental results showed a linear sensitivity of 0.65 pm/ppm [6]. As sensing films, graphene and its derivatives with different architectures, morphologies, and scales have been widely explored for optical fiber gas and vapor sensors [7]. It was also demonstrated recently that a graphene oxide coating layer can significantly enhance the sensitivity of ethanol concentration detection in water based on lossy mode resonance shifts [8]. However, the demodulation scheme of these sensors is in the wavelength region, which always requires high-precision and costly bulk equipment. The temperature cross-sensitivity is also a significant problem for such sensors. In particular, the FBG and microfiber coupler sensors experience very high temperature cross-sensitivity of 0.8 °C [4] and −1450 ppm/°C [6], respectively. On the other hand, the principle of these sensors refers to electronic or structural changes that place with respect to the volume of the coating material [4,5,8,9] or the sensing arm/film [6,10]. Because of the law of diffusion, a longer penetration time is required for inducing significant refractive index changes. Thus, the response time of such sensors varies from ~10 s [6] to several minutes [4,5,9,10].

Compared to the wavelength demodulation system, a photoluminescence (PL) detection system can achieve high-precision measurements with relatively low-cost equipment [11,12,13,14,15,16,17,18]. As one of the most important fluorescent materials, quantum dots (QDs) have outstanding optical luminescent properties compared to conventional organic dyes, including a broad range of absorption wavelengths, a narrow emission spectrum, high quantum yield, robust signal intensity, and high photochemical stability [19,20]. In particular, QDs have high surface-to volume ratios and surface-chemistry-dependent PL properties, namely, they are highly sensitive to various target analytes, such as ions, humidity vapors, and volatile gases. By using different coating technologies and adopting the form of thin films or monolayers, researchers have successfully developed optical fiber sensors based on PL enhancement or quenching which are widely used for temperature [11,12,13], gas [15,16], and chemical ion [17,18] sensing applications. Since the PL intensity is directly correlated with the surface state of the QDs, these sensors always exhibit a fast time response [5]. For example, the 8–10 nm graphene quantum dots formed from multi-walled carbon nanotubes show a fast response time of ~25 s to ammonia gas [18]. The response time of a gas sensor based on PbS colloidal QDs on a poly (ethylene terephthalate) (PET) substrate can be reduced to 4 s [2]. One-dimensional nanofibers with QD-dopant have been demonstrated to be 1–2 orders of magnitude faster than those of RH (relative humidity) or ion sensors based on two-dimensional films or monolayers, with a response time of ~90 ms [15]. However, the fabrication process of this kind of nanofiber sensor is relatively complicated and costly.

In this context, an ethanol vapor sensor based on a microfiber with a QD gel coating is proposed and demonstrated. UV glue was chosen as the gel matrix, which acted as the medium for both hosting the QDs and making ethanol vapor permeate from the surroundings. A simple and low-cost coating system was developed for this study. Exhaustive performance testing was carried out for the fabricated samples, including bending, ethanol sensing, temperature response, and time response tests. Compared with other types of gas sensors, the proposed microfiber ethanol sensor with an easily fabricated robust structure has a much higher sensitivity of −3.3%/ppm, a much lower temperature cross-sensitivity of 0.17 ppm/°C, and a fast response time of 1.1 s. It is believed that the proposed sensor could play a crucial role in real ethanol sensing applications.

## 2. Sensor Fabrication

The schematic diagram of the proposed sensing structure is shown in Figure 1a. The microfiber was coated with a thin layer of CdSe/ZnS QD (average diameter of 10–12 nm, the thickness of the shell is unknown) gel as the sensing film. The QDs were pumped by the 405 nm laser. The fluorescence propagating along the microfiber was recorded. When the sensor was exposed to different concentrations of ethanol vapor, the QD gel reached a fast equilibrium with the atmospheric ethanol vapor, resulting in changes in the output fluorescence intensity.

The microfiber was made from a standard SMF (SMF-28, Corning, Corning, NY, USA) by using a conventional flame-heated taper-drawing technique. The overall taper region was approximately 20 mm and the diameter of the microfiber was approximately 6 μm. The UV glue (NOA 61)-based CdSe/ZnS QDs with a concentration of 1 mg/mL were prepared by Mesolight, Suzhou, China (http://www.mesolight.cc/CdSexZnSxliangzidian.html). The photoluminescence and absorption spectra of the QD gel was tested and the fluorescence emission was at a center wavelength of 530 nm, as is shown in Figure 2a, with quantum efficiency of 80%. The QD gel was coated onto the taper zone of the fiber, using the coating system developed for this study, as is illustrated in Figure 1b. The fiber under coating was first fixed parallel to the linear motor. The tip of the needle with a drop of QD gel then precisely touched the fiber with the help of a microscope. By controlling the moving speed and distance of the linear motor, a uniform and desired coating task was achieved. During our experiment, the coating velocity was set as 0.2 mm/s, and the coating region was approximately 20 mm. After the coating, the QD film was cured by way of UV exposure for approximately 20 min.

## 3. Results

The schematic diagram of the test system is shown in Figure 3. For the spectral test, a 405 nm pump from a laser diode (LD) with a power of 30 MW was launched into the leading fiber as a pump source and the QD fluorescence spectrum was recorded using a micro-optical spectrum analyzer (mOSA, Ocean Optics STS-VIS, Largo, FL, USA) with a resolution of 0.53 nm. The dark field image of the fabricated sensor was first captured. As is shown in Figure 2b, one can see that a uniform green emission light was excited along the coating region. The microscopy images of the typical fabricated sensor are also given in Figure 2c,d. In Figure 2c, the coated (right side) and uncoated (left side) regions are clearly distinguished, and the boundary is also clearly visible. The diameters of the coated and uncoated regions were approximately 7.1 μm and 6.5 μm, respectively, which means that the coating thickness was approximately 0.6 μm. Figure 2d presents the coated region, which shows a smooth coating surface. Overall, the characterization of our fabricated sample confirms a good coating process.

### 3.1. Bending Test

In the subsequent sensing performance experiment, and considering the operation convenience and compact sensor size, we fixed the sensor on a quart slide in a U-shape (see Figure 3). Thus, the bending test was carried out to investigate how the bending factor affected the fluorescence spectrum. The fluorescence spectra under four different bending angles of 176.2°, 117.2°, 86.3°, and 20.3° were recorded, and the dark field images of the corresponding bended sensor were also captured, as is shown in Figure 4. Compared with Figure 4a–d, one can see that under different bending conditions, the excited fluorescence emission keeps uniform along the coated fiber. From Figure 4e, we can see that the four fluorescence spectra keep almost constant. Not only the fluorescence wavelength did not shift, but also the intensity changed very slightly (only 0.022%), which indicates that our fabricated sensor is very robust and bending-insensitive.

### 3.2. Sensing Performance

#### 3.2.1. Ethanol Vapor Sensing

Since ethanol is volatile, the on-slide sensor was sealed in a culture dish at room temperature for alcohol vapor sensing. As is shown in Figure 3, the ethanol with specific quantities was injected into the culture dish, respectively. For each test, the output fluorescence spectrum was recorded when the ethanol fully evaporated and the spectrum remained unchanged. Before the next test, the vapor in the culture dish was exhausted and the fluorescence spectrum returned to its original level. A series of tests was conducted, with ethanol vapor concentrations from 0 to 30.6 ppm by a step of 1.7 ppm (corresponding to 0.5 μL ethanol liquid).

The evolution of the fluorescence spectra of QDs using different ethanol vapor concentrations is presented in Figure 5a. As the alcohol vapor concentration increased, more ethanol vapor molecules diffused into the QD gel, causing the decrease in the refractive index of the QD gel, and finally the fluorescence intensity reduced. To analyze the relationship between the fluorescence intensity and the ethanol vapor concentration, the peak fluorescence intensity at ~530 nm was recorded and normalized. Figure 5b gives the experimental data and linear fitting results. For both the ascending and descending tests, the sensor showed good linear response and the hysteresis was very low. The ethanol sensitivity was calculated as −3.3%/ppm, which is three orders higher than the previous work, which was based on wavelength demodulation methods with a relative sensitivity of 6^−5^%/ppm [6]. It should be noted that the sensitivity can be further enhanced by decreasing the diameter of the microfiber, exploring an optimal QD doping concentration, and fabricating fiber grating pairs on both sides of the microfiber to form a resonant cavity.

#### 3.2.2. Temperature Response

Since the temperature cross-sensitivity always affects sensing performance, the temperature response of the sensor was also explored. The sensor was put into an oven and the temperature was increased from 30 °C to 36 °C by a step of 0.5 °C. The spectrum was recorded at each temperature when it became stable. Figure 6a presents the evolution of the QDs’ fluorescence spectra under different temperatures. As the temperature increased, the fluorescence intensity decreased slightly. The normalized fluorescence intensity as a function of temperature is also given in Figure 6b. By linear fitting, the temperature sensitivity of the sensor was obtained as −0.57%/°C, which leads to a cross-sensitivity of approximately 0.17 ppm/°C. Compared with previous studies [4,5,6], the temperature cross-sensitivity is extremely low. However, for high-precision measurements, temperature controlling should be considered during real applications.

#### 3.2.3. Time Response

Response time is an important property of a sensor. The response time of the reported optical fiber sensors with a similar structure can be tens of seconds to several minutes [4,5,6,7,8,9,10,18]. We measured the response time of the fabricated sensor by real-time recording the QD peak fluorescence intensity at 530 nm when periodically exhausting the ethanol vapor out of the culture dish. The detailed testing process was as follows: a drop of ethanol was injected into the culture dish. Before it totally evaporated, periodical exhausting was conducted. The test results are shown in Figure 7 and one can see that after the exhaustion, due to the sudden decrease in the concentration of the ethanol vapor, the fluorescence intensity increased rapidly. After each exhausting, the dish was sealed again, thus the ethanol continued evaporating and the concentration of the ethanol vapor increased again, causing the fluorescence intensity to decrease. When the intensity reached the original level, the next exhaustion was conducted. From the experimental results, the response and recovery times were calculated at approximately 1.1 s and 1.8 s, respectively, which represents the time the sensor takes to reach 95% and 15% of its final and original value, respectively.

## 4. Conclusions

We have proposed an ethanol sensing structure based on a QD gel-coating microfiber. A batch of samples was prepared using a drawing and coating system developed for this study. The fabricated sensor was first bended to different angles and each fluorescence spectrum was recorded. The test results indicated that our fabricated sensor was very robust and bending-insensitive. Then, the ethanol sensing experiments were conducted, with ethanol vapor concentrations from 0 to 30.6 ppm. The “yo-yo” measurement results showed that the proposed structure had very good repeatability with a linear ethanol response of −3.3%/ppm. Temperature effect was also tested, and the experimental results indicated that the cross-sensitivity was quite low at approximately 0.17 ppm/°C, which can be ignored for normal ethanol vapor sensing applications. Finally, as a key performance parameter, the time response was tested. A fast response time of 1.1 s was obtained, which is much better than the times observed in previously reported studies [4,5,6,7,8,9,10,18]. It is believed that our proposed easily fabricated sensor and its excellent sensing performance could play a crucial role in real ethanol sensing applications.

## Figures and Tables

**Figure 1 sensors-19-00300-f001:**
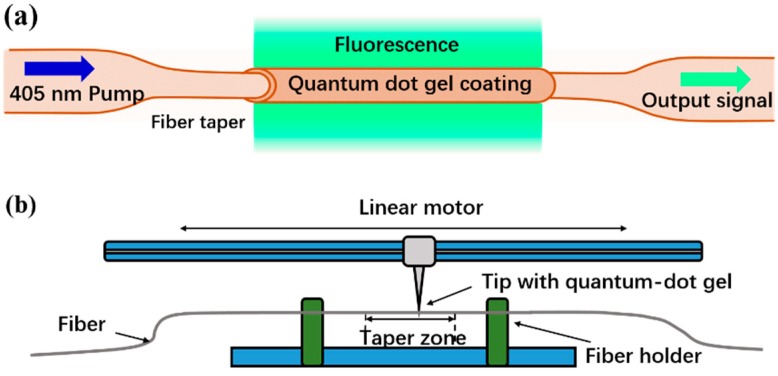
(**a**) The schematic diagram of the RH sensor; (**b**) the schematic diagram of the coating system developed for this study.

**Figure 2 sensors-19-00300-f002:**
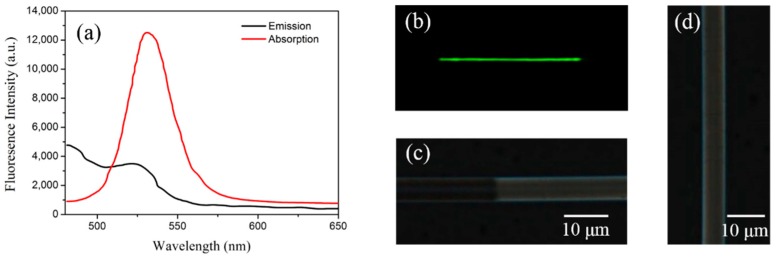
(**a**) The photoluminescence and emission spectra of the QD-UV glue; (**b**) the dark field image of the fabricated sensor under pump; the microscopy image of the fabricated sensor (**c**) with only the right half coated; (**d**) totally coated.

**Figure 3 sensors-19-00300-f003:**
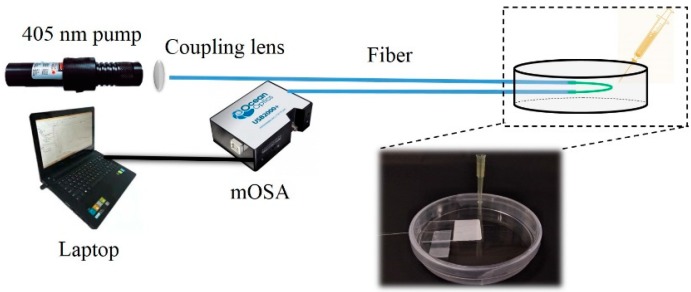
The schematic diagram of the test system.

**Figure 4 sensors-19-00300-f004:**
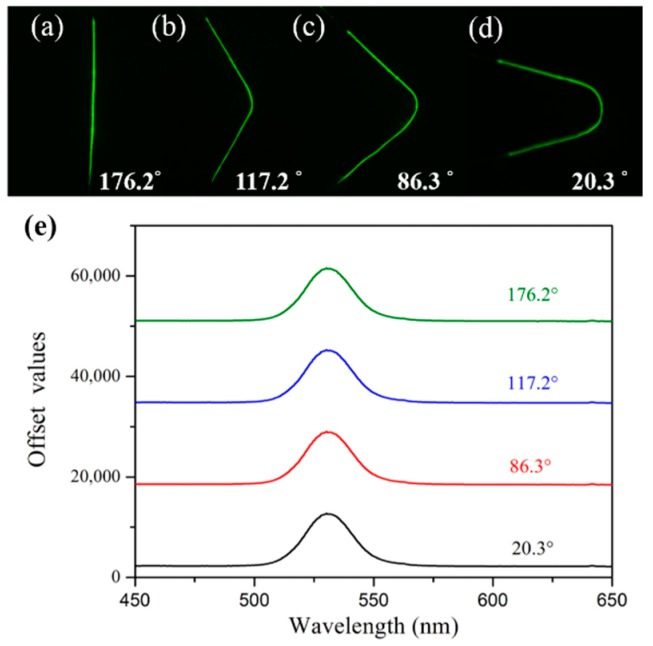
(**a**–**d**) The dark field images of the corresponding bended sensor. (**e**) The PL spectra corresponding to different bending angles.

**Figure 5 sensors-19-00300-f005:**
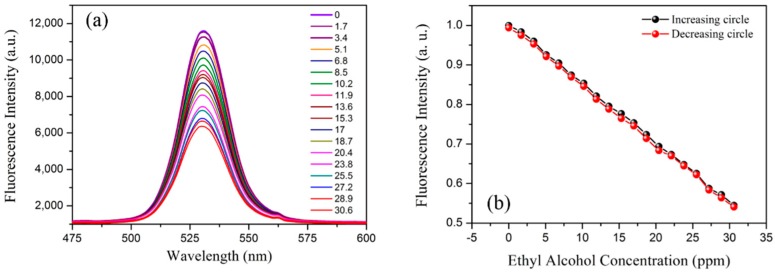
(**a**) The evolution of the fluorescence spectra of QDs using different ethanol vapor concentrations. (**b**) The normalized fluorescence intensity as a function of ethanol concentration.

**Figure 6 sensors-19-00300-f006:**
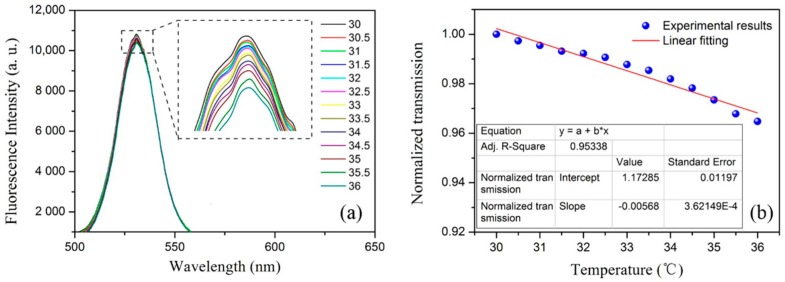
(**a**) The evolution of the QDs’ fluorescence spectra under different temperatures. (**b**) The normalized fluorescence intensity as a function of temperature.

**Figure 7 sensors-19-00300-f007:**
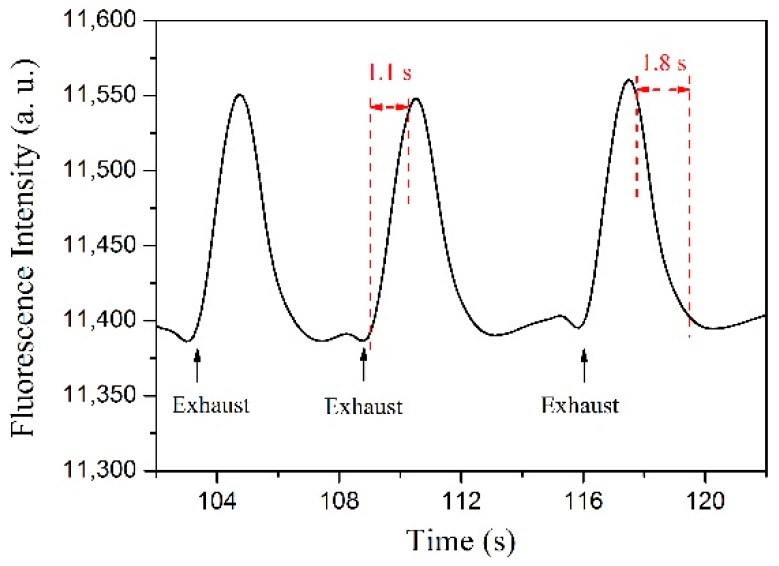
The results of the time response test.

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
