# Peer review of "An Ethanol Vapor Sensor Based on a Microfiber with a Quantum-Dot Gel Coating"

_sensors, 2019, doi:10.3390/s19020300_

Reviewer 1 Report

In this manuscript, Hu et al. present the development and performance testing of an ethanol vapor sensor based on a microfiber with CdSe/ZnS quantum-dots gel coating. This latter includes the bending, ethanol vapor sensing, temperature response, and time response tests. The manuscript is well-written, and experiments are carefully done and well documented. The topic of the manuscript fits the scope of the journal. I suggest acceptance of this manuscript after minor corrections. See comments below:

--- Information about CdSe/ZnS QDs (e.g. average diameter, core radius, outer shell thickness, etc.) should be included briefly in chapter 2, Sensor fabrication, even if the UV glue (NOA 61) based CdSe/ZnS QDs are commercial products.

--- Recent articles considering optical fiber sensors for ethanol detection should also be referenced in the introduction of the manuscript (suggestions: M. Hernaez, A. G. Mayes and S. Melendi-Espina, Graphene Oxide in Lossy Mode Resonance-Based Optical Fiber Sensors for Ethanol Detection, Sensors 2018, 18(1), 58.; T. T. Tung, M. J. Nine, M. Krebsz, T. Pasinszki, C. J. Coghlan, D. N. H. Tran, and D. Losic, Recent Advances in Sensing Applications of Graphene Assemblies and Their Composites. Adv. Funct. Mater. 2017, 27, 1702891).

Author Response

We appreciate and agree with most of the reviewers’ comments. In accordance with your comments, we have made the point-by-point revision. Please find the attachment for details.

Reviewer 2 Report

This paper is suitable for publication in Sensors.

Please, take into account next comments:

- In section 3.3.2, the number of figures must be Figure 6(a) in line 151, and Figure. 6(b) in line 153.

 - In line 163, please review: … decades of seconds …

Author Response

(The authors gave the same response as above.)
